# Reservoir evaporation in a Mediterranean climate: Comparing direct methods in Alqueva Reservoir, Portugal

Carlos Miranda Rodrigues[1,2], Madalena Moreira[1,3], Rita Cabral Guimarães[1,2], and Miguel Potes[4]

[1]MED - Mediterranean Institute for Agriculture, Environment and Development, Pólo da Mitra, Ap. 94, 7006-554 Évora, Portugal.
[2]Department of Rural Engineering, University of Évora, Pólo da Mitra, Ap. 94, 7006-554 Évora, Portugal.
[3]Department of Architecture, University of Évora, Escola dos Leões, Estrada dos Leões, 7000-208 Évora, Portugal.
[4]Institute of Earth Sciences, Institute for Advanced Studies and Research, University of Évora, 7000-671 Évora, Portugal.

**Correspondence:** Rita Cabral Guimarães (rcg@uevora,pt)

**Abstract.**

Alqueva Reservoir is one of the largest artificial lakes in Europe and is a strategic water storage for public supply, irrigation, and energy generation. The reservoir is integrated within the Multipurpose Alqueva Project (MAP), which includes almost 70 reservoirs in a water-scarce region of Portugal. The MAP contributes to sustainability in southern Portugal and has an important impact on the entire country. Evaporation is the key component of water loss from the reservoirs included in the MAP. Evaporation from Alqueva Reservoir has been estimated by indirect methods or pan evaporation measurements, however, specific experimental parameters, such as the pan coefficient were never evaluated. Eddy covariance measurements were performed at the Alqueva Reservoir from June to September in 2014 as this time of the year provides the most representative evaporation volume losses in a Mediterranean climate. This period is also the most important period for irrigated agriculture and is, therefore, the most problematic period of the year in terms of managing the reservoir. The direct pan evaporation approach was first tested, and the results were compared to the eddy covariance evaporation measurements. The total EC evaporation measured from June to September 2014 was 450.1 mm. The mean daily EC evaporation in June, July, August, and September were 3.7, 4.0, 4.5, and 2.5 mm d$^{-1}$, respectively. A pan coefficient, $K_{pan}$, multivariable function was established on a daily scale using the identified governing factors: air temperature, relative humidity, wind speed, and incoming solar radiation. The correlation between the modelled evaporation and the measured EC evaporation had an R$^2$ value of 0.7. The estimated $K_{pan}$ values were 0.59, 0.57, 0.57, and 0.64 in June, July, August, and September, respectively. Consequently, the daily mean reservoir evaporation ($E_{Res}$) was 3.9, 4.2, 4.5, and 2.7 mm d$^{-1}$ for this 4-month period and the total modelled $E_{Res}$ was 455.8 mm. The developed $K_{pan}$ function was validated for the same period in 2017, and yielded an R$^2$ value of 0.68.

This study proposes an applicable method for calculating evaporation based on pan measurements in the Alqueva Reservoir, and it can be used to support regional water management. Moreover, the methodology presented here could be applied to other reservoirs, and the developed equation could act as a first evaluation for the management of other Mediterranean reservoirs.

# 1 Introduction

Reservoirs and water storage are essential in the Mediterranean region for securing urban and industrial water supply, irrigation, and energy generation due to the huge challenges presented by water scarcity in this region (Hoekstra et al., 2012; Alcon et al., 2017; Tomas-Burguera et al., 2017; Rivas-Tabares et al., 2019). Reservoir evaporation is one of the most important components of the water balance, and thus it should be accurately evaluated (Liu et al., 2016). This is particularly important in southern Europe as large investments have been made in irrigation sector here. For instance, in southern Portugal, the Multipurpose Alqueva Project (MAP) with almost 70 reservoirs is the most important example of such investment. The MAP contributes to sustainability in southern Portugal and has an important impact on the entire country. Alqueva Reservoir is the largest surface water reservoir in southern Europe, with a submerged area of 250 km$^2$ and total storage volume of $4150{\times}10^6$ m$^3$ at full capacity. Each 10 mm of evaporation represents a water loss of $2.5{\times}10^6$ m$^3$, which is sufficient to irrigate almost 8.5 km$^2$ of land containing olive trees and, therefore, corresponds to an estimated annual return of 1.1 million euros.

The methodology of Kohli and Frenken (2015), used to estimate evaporation for artificial reservoirs, is based on crop evapotranspiration; it assumes a crop coefficient equal to 1.0, which means that reservoir evaporation is equal to the reference evapotranspiration. Most reservoir managers in the MAP estimate evaporation based on the reference evapotranspiration. Some water system managers use 1000 mm as the reservoir annual evaporation for simplification. In the case of Alqueva Reservoir, with an average reference evapotranspiration of ~1270 mm per year (calculated by the Penman-Montheith method), the evaporation can be $325{\times}10^6$ m$^3$ or 10% of the total usage volume. This means that the local water budget balance has to be well studied to guarantee the sustainability of this important upstream reservoir. An increased accuracy in the evaporation estimation for Alqueva Reservoir is required because of the projected increase in the irrigation area of the MAP and the importance of regional socio-economic development. A previous study on evaporation from Alqueva Reservoir used indirect methods including the energy budget approach, aerodynamic methods, a combination approaches, and a lake model ('FLAKE') (Rodrigues, 2009). This work was based on measurements from a Class A evaporation pan, located in a floating platform in Alqueva Reservoir, between 2002 and 2006, and its comparison with evaporation values obtained by the energy budget approach to establish monthly pan coefficients. However, there has not been a systematic analysis of the governing factors relating to pan evaporation and reservoir evaporation in the Alqueva Reservoir. Accordingly, the current study reports on direct evaporation measurements using eddy covariance (EC) equipment installed on the existing floating platform in the Alqueva Reservoir, which is a part of the framework of the ALEX project (www.alex2014.cge.uevora.pt). Offshore measurements were conducted from June to September 2014, as this is the most representative period of the year for the evaporation volume in a Mediterranean climate, representing ~60% of the total reference evapotranspiration. This period is also very important for irrigation, and is, therefore, the most problematic period of the year for the management of Alqueva Reservoir.

The turbulent fluxes over the water surface, which can be obtained with direct and continuous measurements, evaluate the exchange of water and energy between the surface and the atmosphere (Arya, 2001; Potes et al., 2017). The EC method is usually applied in research because it is a non-invasive technique and provides the most accurate and reliable method for estimating evaporation (Stull, 2001; Allen and Tasumi, 2005; Tanny et al., 2008; Rimmer et al., 2009). The method is

theoretically based on the correlation between the vertical wind speed and air moisture content fluctuation and is a reliable and accurate method to measure open-water evaporation in a location where it is installed (Blanken et al., 2000; Tanny et al., 2008; Nordbo et al., 2011; Richardson et al., 2012; Vesala et al., 2012; Liu et al., 2015; Ning et al., 2015; Ma et al., 2016). However, it requires sophisticated instrumentation that is capable of accurately recording the minimum variations in wind speed, air temperature, and humidity with a high sampling frequency. Furthermore, the equipment is quite expensive and requires continuous maintenance, which means that it is not possible to perform regular measurements. Several studies using EC measurements to evaluate reservoir evaporation have been conducted in various places worldwide (Blanken et al., 2000; Nordbo et al., 2011; Zhang and Liu, 2014; Metzger et al., 2018; Jansen and Teuling, 2020). Another technique to estimate the actual reservoir evaporation based on direct measurements is the pan evaporation method (Riley, 1966). The World Meteorological Organization suggests pan evaporation as the standard method for measuring open-water evaporation (Gangopadhyaya, 1966). However, the relationship between evaporation and meteorological parameters in the pan and in open water bodies differs. Pan measurements generally overestimate evaporation from large water bodies because, in contrast to a lake, a pan receives large quantities of energy through its base and sides, and thus becomes much hotter than a lake. Moreover, the surface area of the water in the pan is much smaller than that of a lake, thus allowing a greater air renewal over the free surface (Jacobs et al., 1998; Lim et al., 2013; Yu et al., 2017). The measured pan evaporation rates are generally 30% higher than that of lake evaporation at the annual scale. The monthly pan coefficients can differ from the commonly used coefficient of 0.7 by more than 100% (Kohler et al., 1955; Linsley et al., 1982; Ferguson et al., 1985). It is expected that the relationship between pan evaporation and lake evaporation should be a function of meteorological parameters, through the modelled $K_{pan}$. The pan evaporation method remains the cheapest and simplest method; hence, this evaporimeter remains the most commonly used instrument to quantify reservoir evaporation. The application of a pan coefficient to convert measured pan evaporation to reservoir evaporation is a method frequently applied in reservoir studies and this pan coefficient could be calculated as a function of meteorological parameters (Allen et al., 1998; Pereira et al., 1995; Pradhan et al., 2013).

The Portuguese public company (Empresa de Desenvolvimento e Infraestruturas do Alqueva - EDIA) that is responsible for the construction and operation of the MAP has a meteorological station with a Class A evaporation pan. The parameterisation of a pan coefficient to convert the measured pan evaporation to reservoir evaporation would provide the MAP with an expeditious reservoir management tool.

Accordingly, the aims of this study were as follows: (i) to evaluate the actual evaporation rates from the Alqueva Reservoir at the EC and Class A pan evaporation locations, and to then analyse their variability with meteorological parameters (i.e. air temperature, relative humidity, wind speed, and radiation); (ii) to estimate the pan coefficient, $K_{pan}$, for the reservoir as an indirect multivariable function and assess the efficiency of pan evaporation in retrieving the evaporation component when EC measurements are unavailable. The study use daily data for the period from June to September 2014, and was validated using data from the same period in 2017.

The paper is organised as follows. Section 2 presents the measurement site, instrumentation, and data. The methodology used in this study is detailed in Section 3, and the results are presented and discussed in Section 4. Finally, Section 5 summarizes the major conclusions.

## 2 Measurement site, instrumentation, and data

### 2.1 Alqueva Reservoir

The Alqueva Reservoir is located within the Guadiana River in Alentejo, southern Portugal (Fig. 1). The reservoir is the largest artificial lake in southern Europe (EDIA, 2020), with an average depth of 16.6 m and a maximum depth of 92.0 m at full capacity. The reservoir has a total capacity of $4150 \times 10^6$ m$^3$ and a water surface area of 250 km$^2$. Alqueva Reservoir is the upstream reservoir of the MAP, which supplies water to approximately 200 000 inhabitants, irrigates 1 200 km$^2$ (will be expanded to 1 650 km$^2$ in the near future), and has an installed hydroelectric power capacity of 530 MW. The Alqueva River basin covers 55 289 km$^2$ and 85% of the area is in Spain. The mean annual precipitation in the Alqueva River basin is less than 550 mm (in the Portuguese area) and the mean annual runoff at the border gauging station (Monte da Vinha station) is 23 mm. At the reservoir, the annual reference evapotranspiration is 1270 mm, as determined by the Food and Agriculture Organization (FAO) Penman–Monteith equation. More than 80% of rainfall occurs between October and April, and during the summer the maximum air temperature ranges on average from 31 ℃ to 35 ℃ (July and August), often reaching values of > 40 ℃. The region is classified as a Csa region according to the Koppen climate classification, which corresponds to a Mediterranean climate (i.e. a temperate climate with dry, hot summers). The summer local time (LT) in Portugal is coordinated universal time (UTC) + 1.

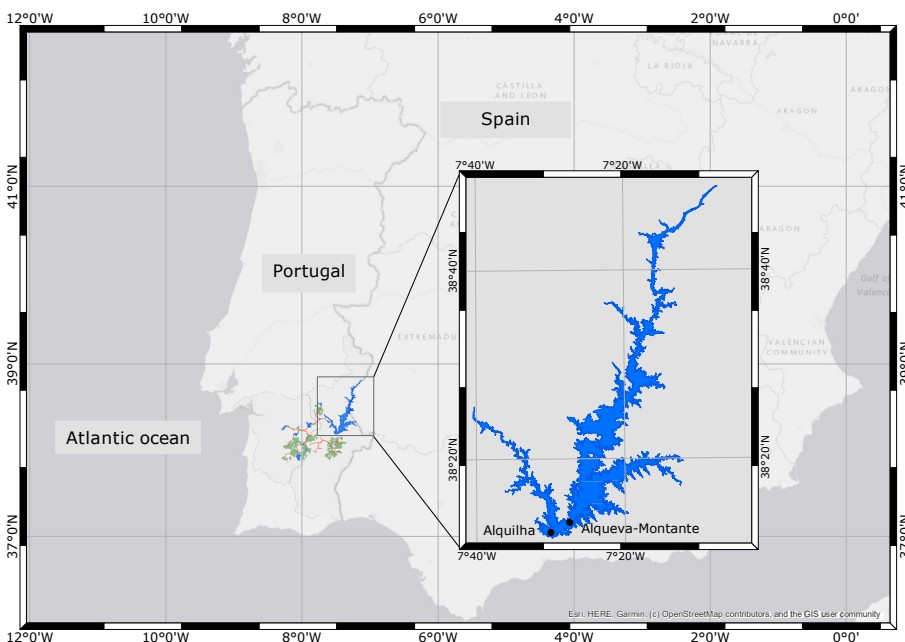

**Figure 1.** Multipurpose Alqueva Project (MAP) location. The expanded map is of Alqueva Reservoir, showing two meteorological stations: Alquilha and Alqueva-Montante.

## 2.2 Instrumentation, data sources, and quality

**Class A pan evaporation**

The Alquilha meteorological station (38° 13' 22.80" N, 07° 30' 03.60" W; elevation of 162 m) is located on the first island upstream of the dam (Fig. 1). The station is part of the environmental monitoring network of the Alqueva Reservoir and is monitored by EDIA, which manages the MAP. The hourly weather variables measured at the station include rainfall (rain-gauge: YOUNG/52202), air temperature and relative humidity (combined sensor: HYDROCLIP), wind speed (3 m above ground) and direction (anemometer and direction sensor: CLIMA), incoming solar radiation (irradiance sensor: IMTSolar/Si-01TCext), and water level readings in a Class A pan (level sensor: Druck/1830). Considering the fact that the station is located on a small island within the reservoir, a very large water fetch upwind of the pan was accounted for this study. The hourly Class A pan evaporation was equal to the hourly level depletion, and accounted for the rainfall effect, and discarded the 3 h period after each refill of the pan. The daily pan evaporation was calculated by considering the starting time water level, the ending time water level, and the upward (water out of the pan) and downward (water into the pan) water level change during a day. The values obtained when the water level in the pan was below a threshold value (10 cm), according to Allen et al. (1998) and WMO (2018), were discarded. Anomalous values were also discarded. For the study period (June to September 2014), 18% and 15% of the data were discarded at hourly and daily scales, respectively, during the quality control process. Discarded and missing data were filled with the average value calculated for the study period (Jun-Sep).

**Eddy covariance system**

Alqueva-Montante (38° 13' 24.75" N, 07° 27' 34.18" W) meteorological and hydrologic station (Fig. 1) is part of the Portugal Network for Water Resource Monitoring (https://snirh.apambiente.pt). The measuring equipment is installed on a floating platform to measure air temperature, relative humidity, wind speed/direction, downward radiation, pressure, and precipitation. These parameters (except for precipitation as this is accumulated during a given period) are measured at a frequency of one value per minute, while averages are calculated for 30 min. The weather station also measures the reservoir water temperature and water quality parameters, which are not used in the present study. The maximum water depth is ∼65 m at the station site, and the shore distance is greater than 300 m; however, these values vary slightly with the type of platform anchorage (i.e. by ropes tied to three sunken blocks), thus allowing longitudinal displacements and rotation on itself.

Within the framework of the ALEX project (www.alex2014.cge.uevora.pt), this instrumented floating platform was equipped with one EC system—an integrated open path $CO_2$/$H_2O$ gas analyser and a 3D sonic anemometer (IRGASON; Campbell Scientific)—at a height of 2 m above the reservoir surface. The variables measured by the IRGASON were $u$, $v$ and $w$ components of wind speed, sonic temperature (computed from the measured sound speed), $H_2O$ and $CO_2$ concentration, and sonic anemometer and gas analyser quality flags. Data were sampled at 20 Hz and the filter time delay was 200 ms (Potes et al., 2017). Turbulent time-series were linearly detrended and a double-axis rotation was applied to the wind speed components. The turbulent fluxes of momentum, heat, and mass ($H_2O$) were calculated as 30 min covariances between the fluctuations of the vertical wind component ($w$), temperature, and the $H_2O$ concentration, respectively. The air density fluctuations were corrected for thermal expansion and water vapour dilution, and the sonic temperature was corrected for humidity. These cor-

140 rections were, then, applied to the flux calculations (Potes et al., 2017). Furthermore, data quality criteria and filters were applied for the study period. Approximately 3% of the original data was discarded based on i) a signal strength (from the gas analyser) of < 0.7, ii) footprints (fetch) with values of X90 of > 300 m, and iii) all data leading to negative values for the $H_2O$ covariances resulting in negative latent heat (evaporation) fluxes. Discarded data was filled with the average value calculated for the study period (Jun-Sep).The predominant wind direction was between 210º and 360º (68% with 30 min resolution), and

145 97% of the mean speed wind measurements (with 30 min resolution) was < 6 ms$^{-1}$ (Fig. 2). In order to assess for the possible contamination for the floating platform on the EC evaporation measurement, two wind direction filters (having as reference the EC system orientation) were applied to flux data. The two filters considered (Evap_fill180 and Evap_fill100) were from wind directions between 90 and 270 degrees and 130 and 230 degrees, as they represent winds that pass through the platform before reaching the EC instrument. To understand the impact of applying a filter of wind direction on the EC evaporation dataset, a

150 comparison was made between the daily cycle without any wind direction filter and with a wind direction filter of i) 180º and ii) 100º (Fig. 3a). The correlations between the daily cycle with a 180º filter and without a filter ($R^2$ = 0.985) and between the daily cycle with a 100º filter and without a filter ($R^2$ = 0.993) are presented in Fig. 3b and 3c. By analysing these figures, we can conclude that the platform does not affect the flux data, according to the wind direction.

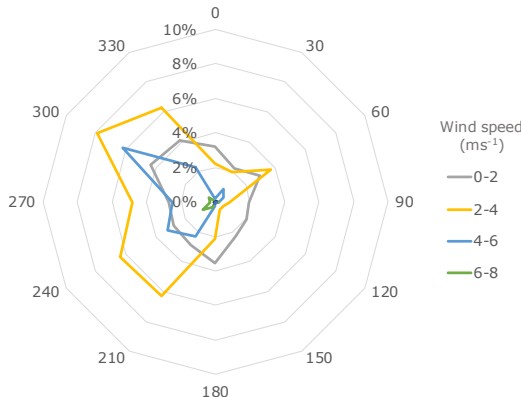

**Figure 2.** Wind rose for Alqueva-Montante meteorological station from June to September 2014.

## 3   Methodology

This section describes the methodology used to estimate evaporation from the Alqueva Reservoir based on the measurements taken at Alquilha station. It is proposed that the actual evaporation from the reservoir could be estimated using the relationship between the Class A pan evaporation measurements (at Alquilha station) and a pan coefficient multivariable function, as determined by Allen et al. (1998), but for reference evapotranspiration. Although the conditions surrounding a site can influence the pan coefficient, this aspect is not considered here as the fetch in the wind direction was irrelevant, as mentioned in Section

2.2. Processed data of pan and EC evaporation (Section 2.2) were used to develop a multivariable pan function.

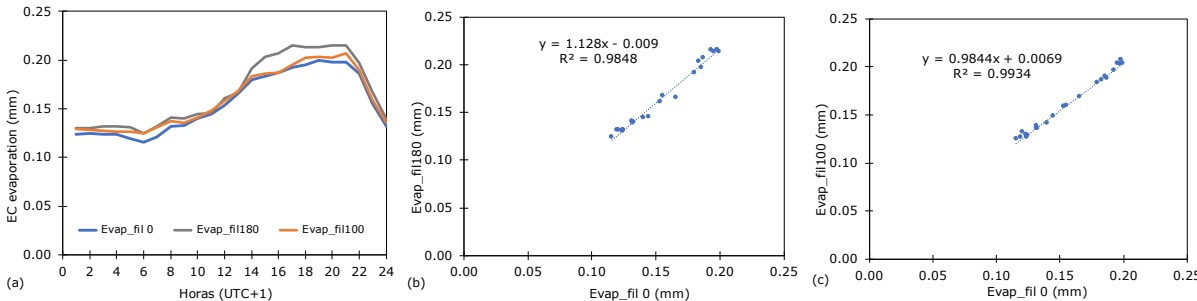

**Figure 3.** (a) Daily cycle of the eddy covariance (EC) evaporation ($E_{EC}$) with and without wind direction filters; (b) correlation between the EC evaporation with a 180º wind direction filter ('Evap_fil180') and without the filter ('Evap_fil 0'); (c) correlation between the EC evaporation with a 100º wind direction filter ('Evap_fil100') and without the filter ('Evap_fil 0'), for Alqueva-Montante station from June to September 2014.

First, relationships between the EC measurements and meteorological parameters (air temperature, relative humidity, wind speed, and solar radiation) measured at the Alqueva-Montante station were determined. These four meteorological parameters were selected primarily because they are the factors governing evaporation, as usually described in the literature (e. g., Allen et al., 1998), and are the parameters measured in the Alquilha meteorological station. The daily cycle of evaporation and normalised meteorological parameters were analysed to assess their behaviours during the day. A sensitive analysis at the hourly scale was also performed for the factors governing evaporation from the Alqueva Reservoir.

Second, the relationships between pan evaporation measurements and the same meteorological parameters, but as measured at Alquilha station (at hourly and daily scales), were determined.

Third, the correlation between EC evaporation and pan evaporation was determined and the daily cycles of the normalised pan evaporation and normalised EC evaporation were compared.

Fourth, a sensitivity analysis was performed, calculating the correlation of the daily pan evaporation and daily EC evaporation with air temperature, relative humidity, windspeed, and solar radiation.

Fifth, the daily multivariable pan coefficient series was calculated by dividing the daily values of EC evaporation with the corresponding daily values of pan evaporation.

Sixth, a function was fitted to this series based on the physical relationships among the different meteorological parameters measured at the Alquilha station (at the daily scale). Several functions were attempted, and the one with the best determination coefficient ($R^2$) was chosen. To determine the optimal parameter estimates, the Generalized Reduced Gradient (GRG) method (Lasdon et al., 1974) was used with the aid of the Excel solver tool. The best parameter estimates were those that minimised the residual sum of squares.

# 4   Results and discussion

## 4.1   Eddy covariance evaporation

The total EC evaporation measured from June to September 2014 was 450.1 mm. The mean daily EC evaporation in June, July, August, and September were 3.7, 4.0, 4.5, and 2.5 mm d$^{-1}$, respectively. The correlations between the hourly EC evaporation and wind speed, air temperature, relative humidity, and incoming solar radiation are presented in Fig. 4. At the hourly scale, a positive correlation was observed between the EC evaporation and i) wind speed ($R^2$ = 0.50) and ii) air temperature ($R^2$ = 0.20), whereas a negative correlation was observed between open evaporation and relative humidity ($R^2$ = 0.30). There was no correlation between open water evaporation and incoming solar radiation.

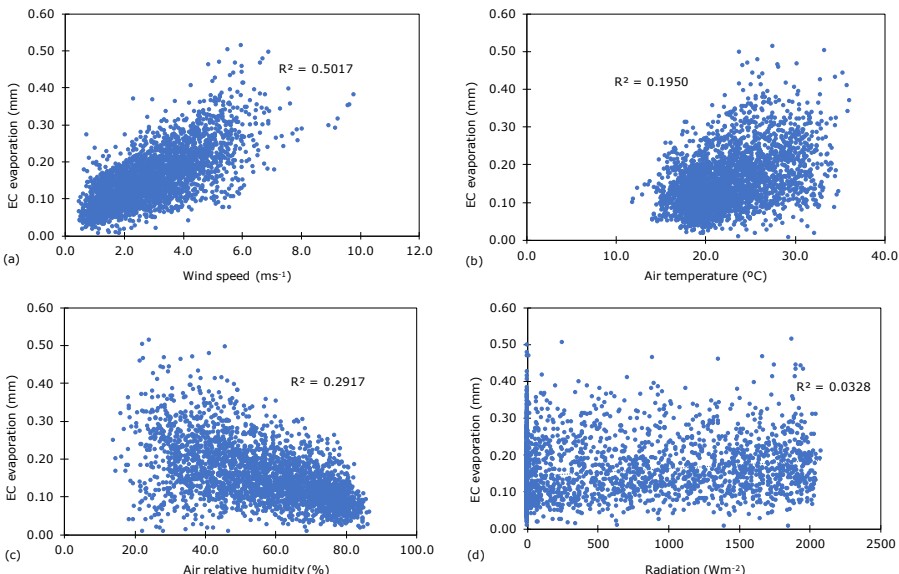

**Figure 4.** Hourly correlation between the eddy covariance (EC) evaporation (E$_{EC}$) and (a) wind speed ($U$), (b) air temperature ($Ta$), (c) relative humidity ($RH$) of air, and (d) solar radiation ($Rad$) at Alqueva-Montante station.

The daily cycles of evaporation and the meteorological parameters allow the variation during an average day to be analysed. The normalisation of the mean values of the meteorological parameters was performed to unify the scale of the parameters. The daily cycle of evaporation and the four normalised meteorological parameters measured at the Alqueva-Montante station are presented in Fig. 5. As expected, the air temperature and relative humidity exhibited opposite trends. There was a slight variation in the wind speed during the morning and a considerable increase after 10:00 LT, which induced a variation in evaporation. After 6:00 LT, evaporation increased continuously until 21:00 LT, along with increases in radiation and wind speed but decreasing relative humidity. Incoming solar radiation contributed to evaporation with a delay that could be explained by the variation in the energy stored in the water column. The increase in solar radiation may lead to an increase in the stored energy in the water column (Potes et al., 2017; Nordbo et al., 2011). The air temperature subsequently decreased compared

to the water temperature, and the energy was released into the air, thereby increasing evaporation. An evaporation inflexion point occurred at 14:00 LT when the incoming solar radiation began to decrease. Accordingly, evaporation began to decrease at 21:00 LT when there was no solar radiation.

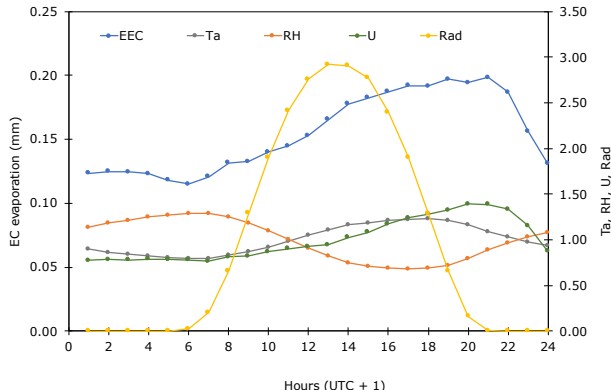

**Figure 5.** Mean daily cycle of the eddy covariance (EC) evaporation ($E_{EC}$) (left y-axis) and normalised air temperature ($Ta$), relative humidity ($RH$) of air, wind speed ($U$), and solar radiation ($Rad$) (right y-axis) from June to September 2014 at Alqueva-Montante station.

## 4.2 Class A pan evaporation

The total pan evaporation measured from June to September 2014 was 797.9 mm. The mean daily pan evaporation in June, July, August, and September were 6.9, 7.7, 7.3, and 4.3 mm d$^{-1}$, respectively.

Such as for the EC evaporation, a positive correlation was observed between the hourly pan evaporation and air temperature ($R^2$ = 0.55), whereas a negative correlation was found between the hourly pan evaporation and relative humidity ($R^2$ = 0.53). In contrast, a positive correlation was observed between the hourly pan evaporation and incoming solar radiation ($R^2$ = 0.35), and a weak positive correlation was evident between the hourly pan evaporation and wind speed ($R^2$ = 0.05). The daily cycle of evaporation and the four normalised meteorological parameters (wind speed, air temperature, relative humidity, and solar radiation) measured at Alquilha station are presented in Fig. 6. In the morning period, the solar radiation begins at 8:00 LT and with that an increase in air temperature and a decrease in relative humidity. At 11:00 LT wind speed starts to increase and around 12:00 LT occurs the trigger of the evaporation pan. The trend of the pan evaporation followed the trend of solar radiation but with a delay of about 3 hours, whereby the maximum value was at 16:00 LT when the relative humidity was at the minimum. Pan evaporation reduced as the air relative humidity increased.

## 4.3 Correlation between EC evaporation and pan evaporation

The correlation between daily eddy covariance evaporation and daily pan evaporation was determined for the study period (June-September) and is shown in Fig. 7. Fig. 7a shows a poor linear correlation between the EC evaporation and pan evaporation during the entire study period ($R^2$ = 0.37). This was also the case when observing the plots for each month:$R^2$ = 0.1882 in

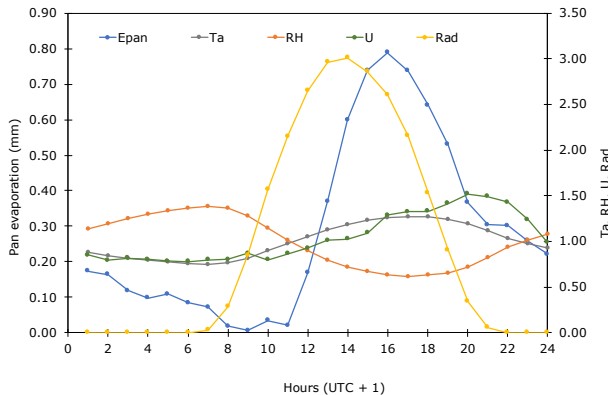

**Figure 6.** Mean daily cycle of pan evaporation ($E_{pan}$) (left y-axis) and normalised air temperature ($Ta$), relative humidity ($RH$) of air, wind speed ($U$), and solar radiation ($Rad$) (right y-axis) from June to September 2014 at Alquilha station.

June (Fig. 7b), $R^2$ = 0.0458 in July (Fig. 7c), $R^2$ = 0.3345 in August (Fig. 7d), and $R^2$ = 0.4693 in September (Fig. 7e). These results shows that the relationship between both evaporation could not be considered linear and reveal the importance of finding a nonlinear function to correlate EC evaporation and pan evaporation. The daily cycles of the normalised pan evaporation and normalised EC evaporation are compared in Fig. 8. The two evaporations exhibited different behaviours; pan evaporation varied widely over the day, with zero evaporation at 9:00 LT and the maximum at 16:00 LT. The maximum mean daily pan evaporation was 2.75-fold that of the mean daily value. In contrast, the daily cycle of the EC evaporation fluctuated comparatively little over the day. During the night and early morning, EC evaporation was ∼80% of the daily mean value, with the minimum at 6:00 LT. During the late afternoon, the EC evaporation increased due to the increased wind speed (Fig. 5). The maximum daily mean evaporation occurred at 21:00 LT and it was 125% of the daily mean value.

These results agree with a previous study by (Salgado and Le Moigne, 2010) for the same reservoir, wherein the authors observed an absolute minimum and maximum at 6:00 LT and 21:00 LT, respectively. Although both types of evaporation measurements used similar times for calculating the mean daily value (between 12:00 LT and 13:00 LT), the significant dissimilarities over the day resulted from the large difference between the size of the pan and the size of the reservoir as these may lead to different heat storage capacities. Owing to the reduced water height in the pan, the amount of energy it would have received through radiation and conduction through the walls of the pan is incomparably higher than that received by the reservoir water. Moreover, the reduced area of the pan would have tended to enhance the loss of water through evaporation because it facilitates the removal of air-saturated layers at the water–air interface.

### 4.4 Sensitivity analysis of pan evaporation and EC evaporation versus meteorological variables

A sensitivity analysis of the daily pan evaporation and daily EC evaporation with air temperature, relative humidity, wind speed, and solar radiation, was carried out and the results are presented in Fig. 9. Fig. 9a shows a non-linear correlation between evaporation (EC and pan evaporation) and wind speed. It can be observed that both evaporations have a positive linear

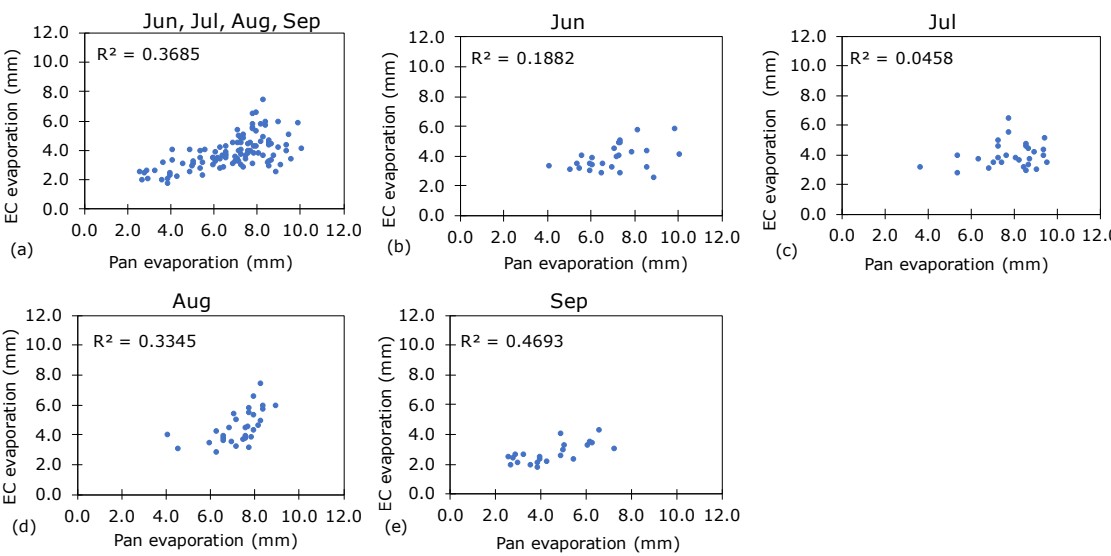

**Figure 7.** Correlation between the daily eddy covariance (EC) evaporation ($E_{EC}$) and the daily pan evaporation($E_{pan}$): (a) June to September 2014; (b) June 2014; (c) July 2014; (d) August 2014; (e) September 2014.

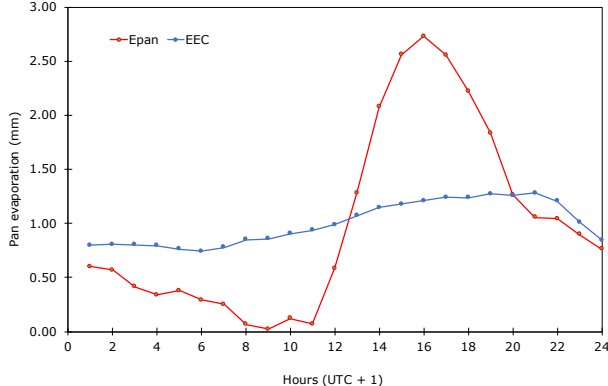

**Figure 8.** Mean daily cycle of the normalised pan evaporation ($E_{pan}$) and the eddy covariance (EC) evaporation ($E_{EC}$).

correlations with air temperature (Fig. 9b) and radiation (Fig. 9d). Fig 9c shows a negative correlation between evaporation and air relative humidity. The value of $R^2$ of pan evaporation with air temperature, air relative humidity, and radiation is greater

than the $R^2$ of the EC evaporation with the same parameters. In contrast the $R^2$ of EC evaporation with wind speed is greater than the pan evaporation with the wind speed parameter.

Based on this sensitivity analysis, it was inferred that the four parameters influence both EC evaporation and pan evaporation, and strengthen the ability to establish a relationship between the open EC evaporation and pan evaporation on a daily scale as discussed in Section 4.5.

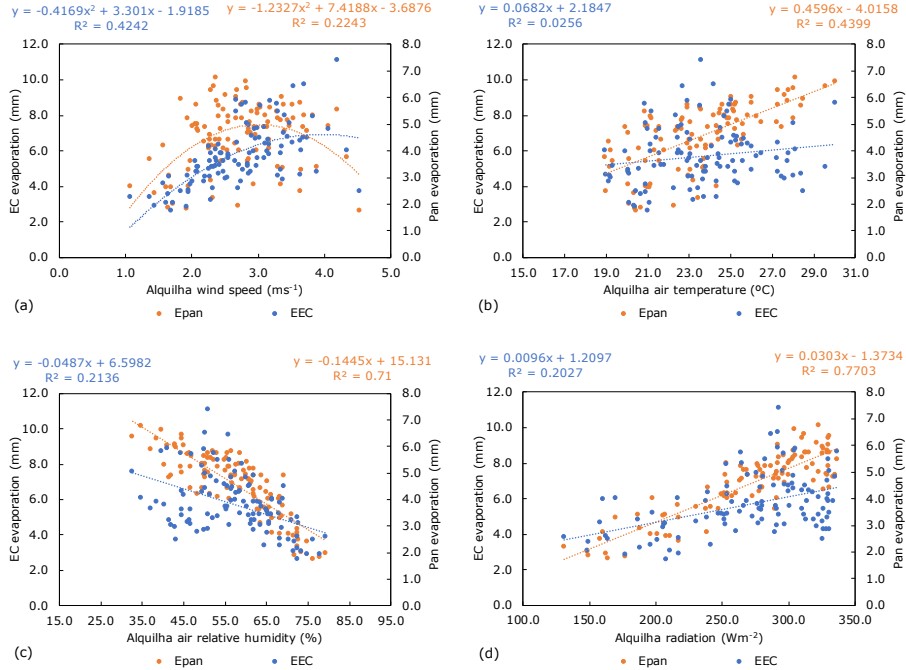

**Figure 9.** Sensitivity analysis of the daily eddy covariance (EC) evaporation ($E_{EC}$) and the daily pan evaporation ($E_{pan}$) from June to September 2014, with (a) wind speed; (b) air temperature; (c) relative humidity of air; (d) solar radiation.

### 4.5 Pan evaporation coefficient model

The pan evaporation coefficient ($K_{pan}$) was calculated as a function of the four meteorological parameters measured at the Alquilha station because this station will be used in the future to obtain data to support water management and decision-making. Consequently, the reservoir evaporation ($E_{Res}$) is estimated by multiplying the Alquilha Class A pan evaporation $E_{(pan)}$ measurement (at Alquilha) with the modelled $K_{pan}$.

The pan evaporation coefficient model was expressed by a multivariable function as shown in Eq. (1):

$$K_{pan} = aU + bTa + cLN(RH) + dLN(Rad) + eTaLN(Rad) + f \tag{1}$$

where $a$, $b$, $c$, $d$, $e$, and $f$ are specific constants; $U$ is the average daily wind speed at a height of 2 m at the Alquilha station (m s$^{-1}$); $Ta$ is the average daily temperature at Alquilha station (ºC); $RH$ is the average daily relative humidity at Alquilha station (%); and $Rad$ is the total daily radiation at Alquilha station (W m$^{-2}$).

By using an objective function to minimise the residual sum of squares, the parameterisation of the specific constants was performed by optimisation using the GRG method; thus, Eq. (1) becomes:

$$K_{pan} = 0.0925U + 0.1531Ta - 0.2558LN(RH) + 0.2593LN(Rad) - 0.0308TaLN(Rad) + 0.3489 \qquad (2)$$

The daily mean modelled $K_{pan}$ was 0.59, 0.57, 0.57, and 0.64 for June, July, August, and September, respectively. These values are slightly larger than those obtained directly by the ratio of the EC evaporation to pan evaporation (0.54). Rodrigues (2009) reported monthly $K_{pan}$ values between 0.70 and 0.90 for the same summer period and reservoir but using a different approach; he estimated $K_{pan}$ values by relating pan evaporation, measured from a floating pan at the Alqueba-Montante platform, and reservoir evaporation obtained by the energy budget approach.

Fig. 10 presents $E_{Res}$ determined from the pan evaporation coefficient model and the measured EC evaporation. The $R^2$ value of 0.74 indicates that this model can estimate the $E_{Res}$ quite well. The total modelled $E_{Res}$ for the period from June-September was 455.8 mm, which corresponds to 101.3% of the EC evaporation and 76% of the site reference evapotranspiration calculated by the Penman–Monteith equation (Allen et al., 1998). The modelled daily mean $E_{Res}$ in June, July, August, and September was 3.9, 4.2, 4.5, and 2.7 mm d$^{-1}$, respectively.

The ability of the model was tested for the period from June-September 2017 (Fig. 11; $R^2$ = 0.68); thus, the model could estimate the $E_{Res}$ despite high measured evaporation rates and a reduced number of available daily pan evaporation measurements.

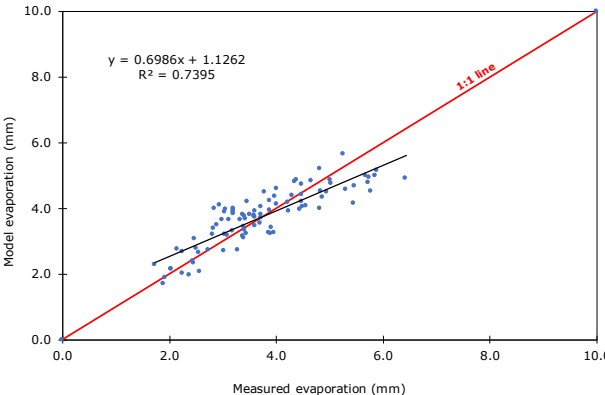

**Figure 10.** Modelled daily evaporation ($E_{Res}$) versus measured daily evaporation ($E_{EC}$) from June to September 2014.

## 5  Conclusions

The study aimed to develop a method to evaluate the evaporation from Alqueva Reservoir, located south-east of Portugal, based on Class A pan measurements, thus providing an evaluation tool for water management within the Multipurpose Alqueva Project (MAP) and for other reservoirs with a Mediterranean climate.

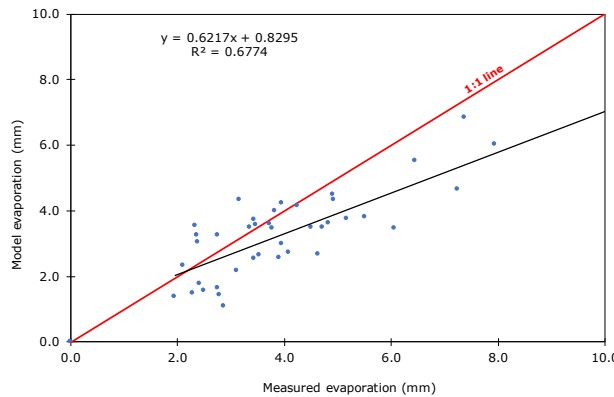

**Figure 11.** Modelled daily evaporation ($E_{Res}$) versus measured daily evaporation ($E_{EC}$) from June to September 2017.

Water fluxes were continuously measured from June to September 2014 using the EC method at the Alqueva-Montante station to obtain accurate reservoir evaporation measurements. The total EC reservoir evaporation from June to September 2014 was 450.1 mm, and the mean daily evaporation in June, July, August, and September were 3.7, 4.0, 4.5, and 2.5 mm d$^{-1}$, respectively. Considering the most important atmospheric factors controlling evaporation, a positive correlation between the EC evaporation, wind speed, and air temperature, a negative correlation for the relative humidity, and no correlation between

EC evaporation and solar radiation was observed at an hourly scale.

    The Class A pan installed at the Alquilha station provided hourly and daily pan evaporation values. The total pan evaporation from June to September 2014 was 797.9 mm, and the mean daily evaporation in June, July, August, and September were 6.9, 7.7, 7.3, and 4.3 mm d$^{-1}$, respectively. Positive correlations were observed between the hourly pan evaporation and air temperature and solar radiation, whereas a negative correlation was found between the hourly pan evaporation and the relative

humidity. A weak correlation existed between the hourly pan evaporation and wind speed.

    A sensitivity analysis of the daily pan evaporation and daily EC evaporation with air temperature, relative humidity, wind speed, and solar radiation, strengthen the ability to establish a relationship between the open EC evaporation and pan evaporation at the daily scale.

    We found that the daily pan evaporation coefficient could be expressed by a multivariable function of wind speed, air

temperature, relative humidity, and solar radiation measured at Alquilha station. Further, model validation was performed for the same four summer months in 2017. The modelled pan coefficients ($K_{pan}$) were 0.59, 0.57, 0.57, and 0.64 in June, July, August, and September, respectively; the modelled daily mean $E_{Res}$ was 3.9, 4.2, 4.5, and 2.7 mm d$^{-1}$ for June, July, August, and September, respectively. The total modelled evaporation was 455.8 mm, remarkably similar to the total output from EC measurements, and corresponds to 101.3% of the measured EC evaporation from the reservoir.

The evaporation model proposed in this study can assist and improve water management in the MAP. Moreover, the methodology could also be applied to other reservoirs, and the equation developed for Alqueva Reservoir could act as a first evaluation for the management of other reservoirs in the region.

*Author contributions.* The four authors conceptualised the study. CMR and MP designed and carried out the experiments. RCG performed the model simulations. MM wrote the first draft manuscript. All the four authors contribute to the analysis, interpretation and writing.

*Competing interests.* The authors declare that they have no conflict of interest

*Acknowledgements.* This work is funded by National Funds through the Foundation for Science and Technology (FCT) under Project UIDB/05183/2020, Project ALEX 2014 (EXPL/GEO-MET/1422/2013), Project ALOP (ALT20-03-0145-FEDER-000004), and AGIR (PDR2020-1.0.1-FEADER-031864).

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
