# Peer review of "Reservoir evaporation in a Mediterranean climate: Comparing direct methods in Alqueva Reservoir, Portugal"

_Hydrology and Earth System Sciences, 2020_

## Referee Comment (RC1) · Femke Jansen (Referee) · 13 Aug 2020

The manuscript by Rodrigues *et al.* provides an interesting and clear analysis of evaporation from a large reservoir where two different direct measurement methods are compared. By establishing a pan coefficient that is a function of identified governing factors is was shown that on a daily timescale pan measurements can be used to resemble the evaporation rates measured by an eddy covariance installation. The study is straightforward and concise, and the results presented follow directly from the aims and hypotheses of the study. In general, the study provides a good base for further exploration of total reservoir evaporation in the Mediterranean climate using simple measurement instruments, and with that provide information to support the management of the reservoir. With this I think that this study is interesting for publication. However, I would like to suggest the need of minor revisions, which is mainly related to the structure of the manuscript, and the reproducibility of the study.

General comments

In terms of reproducibility of the study, I think that in general the description of some of the methods that are applied are not sufficient. This includes:
- The quality control process of the Class A pan in section 2.2. Please elaborate on what method this quality control process is based on.
- The sensitivity analysis - I would like to read more on how the authors have performed the sensitivity analysis. This does not become clear from the Methodology section, nor from the results in section 4.4.
- The factors governing evaporation – it needs more clarity on how the factors governing evaporation were determined. These factors are mentioned in the Methodology section, and form the base of the pan coefficient function that is developed. Are these governing factors identified based on literature or other results that are not shown here?
- The multivariable nonlinear pan coefficient function - could the authors explain how they came to the form of the multivariable nonlinear pan coefficient function, apart from the explanation that a linear function would not describe the correlation between EC evaporation and pan evaporation well.

A clear description of the figures that are presented as results is sometimes lacking in my opinion. This is the case for figures 3, 7 and 9. What do we see in this figure, how do we read it, what is the main message that the reader can take from it? I think this will help your story to come across more direct and focussed, and will improve the guidance of the reader towards the conclusions that are well supported by the results.

Another general comment that I would to make is to see if a better balance can be achieved between the size of the sections. Sections 3 and 4.4 are relatively short and misses information. Probably this can already be improved by applying the two comments mentioned above.

The conclusions section somewhat misses a concluding statement and is now presented more as a summary. Furthermore, some new numbers are shown in this section, which is not the appropriate place to present new results. Referring to p.13 l.253.

Specific comments

p.2 lines 46-48; how was the total reference evapotranspiration calculated? Using Penman-Monteith as mentioned at p.12 line 236?

p.3 lines 77-82; I would like to suggest to describe at what timescales the study focusses.

p.5 lines 134/135; please check if the negative latent heat fluxes found are indeed erroneous, or is there condensation happening?

p.5 lines 137-141; it does not become clear how the authors have applied this filter. Does the wind direction filter have a range of 180° and 100° respectively, or is there a filter from 180° and 100° towards 360°? Please clarify from which to which wind direction the filter is applied.

p.7 line 146/147; What conditions surrounding a site can influence the pan coefficient? Could the authors further explain if indeed those conditions can be ignored because the fetch in the wind direction was found not to be relevant.

p.7 line 161; How did the authors deal with the data that was filtered out in calculating the total evaporation amount?

p.7 line 173/174; The authors mention that the delay of evaporation is related to the variation in the energy storage in the water body, however this is not shown in figure 5. Do the authors have data on this that could be presented?

p.7 lines 174/175; I think this argumentation could be written down more clearly. The increase in energy storage in the water body by solar radiation is not depending on the gradient of air-water temperature. The solar radiation will penetrate the water surface in any case.

p.7 lines 176-178; at line 166 it is presented that there no correlation was found between open water evaporation and incoming solar radiation. However, in line 176-178 it is presented as if there is a direct correlation between the variables. Please elaborate.

p.10 lines 205-207; which results support this statement? As far as I can see there is no data presented on heat storage.

p. 12 lines 243/244; it would be interesting to know whether the method presented in this study can indeed be applied to other reservoirs with a Mediterranean climate. Could the authors discuss further on this?

Technical corrections
p.2 line 28, 29; not sure if $hm^3$ is a common unit to use. Consider changing.
p.2 line 31; (Kohli and Frenken, 2015) -> Kohli and Frenken (2015)
p.2 lines 55/56; This sentence seems not in the right place in this location in the paragraph. Consider bringing it forward.
p.3 line 59; add 'it' to the sentence: 'which means that **it** is not possible…'
p.3 line 64; waterbodies -> water bodies
p.3 lines 83-85; This paragraph might be redundant. Especially mentioning about section 1, which the reader at that moment has just read.
p.7 line 164 and other lines; trend should be correlation?
p.7 line 166; open evaporation -> open water evaporation
p.8 line 187; The most importance differences with what?
p.8 lines 187/188; The dominance of wind speed over solar radiation in relation to open water evaporation? Please clarify.
p.12 line 230; (Rodrigues, 2009) -> Rodrigues (2009)

p.12 line 239; please clarify what is meant with 'high measured evaporation'? High evaporation rates? High measurement frequency?

p.13 line 257; significative -> significant. Or should it be 'weak' instead of 'no significant' following from section 4.2.

---

## Author Comment (AC1) · 8 Sep 2020

[hess,manuscript]copernicus

Dear Femke Jansen

Manuscript reference No. HESS-2020-283

We would like to thank you, for your insightful comments, which unquestionably contributed to improve our manuscript. We believe that we were able to fully and ade-

quately respond and address all your questions and recommendations by re-writing important sections of the manuscript. In the following pages are our point-by-point responses to each of your comments as well as your own comments.

Revisions in the text are shown using green colour font for [example] additions , and strike through red font [example] for deletions.

**General comments**
**- The quality control process of the Class A pan in section 2.2. Please elaborate on what method this quality control process is based on**.

The quality control process is based on the analysis of the existing data in order to discard the values that, for any reasons, could not be considered adequate. Following, we have discarded:

- The values obtained 3 hours after each refill of the pan;

- The values obtained when the water level in the pan is below a threshold value (10 cm), according to Allen et al., 1998 and WMO, 2018;

- The anomalous values.

We have provided more details for description the quality control process in section 2.2:

p5 line 110;

"...The daily pan evaporation was calculated by considering the starting time water-level, the ending time water-level, and the upward (water out of the pan) and downward (water into the pan) water level change during a day. The values obtained when the water level in the pan is below a threshold value (10 cm), according to Allen et al. (1998)

and WMO (2018), was discarded. It was also discarded the anomalous values. For the study period (June to September 2014), 18% and 15% of the data was discarded at hourly and daily scales, respectively, during the quality control process."

**- The sensitivity analysis - I would like to read more on how the authors have performed the sensitivity analysis. This does not become clear from the Methodology section, nor from the results in section 4.4.**

The sensitivity analysis was done by determining the correlation between evaporation (daily pan evaporation and daily EC evaporation) and the four meteorological parameters measured at Alquilha station (because this station will be used to obtain data in the future).

We add some additional text to the manuscript in order to explain more clearly the sensitivity analysis in section 4.4. We add also some text in Methodology and in Conclusions.

The text is now as follows: p7 line 151;

"...Second, the relationships were determined between pan evaporation measurements and the same meteorological parameters, but as measured at Alquilha station (at hourly and daily scales).

Third the correlation between EC evaporation and pan evaporation were determined and the daily cycles of the normalised pan evaporation and normalised EC evaporation are compared.

Forth a sensitivity analysis of pan evaporation and EC evaporation versus meteorological variables were performed...."

p10 line 212;

"A sensitivity analysis of the daily pan evaporation and daily EC evaporation with air temperature, relative humidity, wind speed, and solar radiation, was carried out and the results are presented in Fig.9. Fig. 9a show a non-linear correlation between evaporation (EC and pan evaporation) with wind speed. It can be seen that both evaporations have a positive linear correlation with air temperature, Fig. 9b, and radiation, Fig. 9d. In Fig 9c it can be seen a negative correlation between evaporation and air relative humidity. The value of R2 of pan evaporation with air temperature, air relative humidity and radiation is greater than the R2 of the EC evaporation with the same parameters. On the contrary the R2 of EC evaporation with wind speed is greater than the pan evaporation with the wind speed parameter. Based on this sensitivity analysis, the four parameters appear to cause influence in both EC evaporation and pan evaporation, and strengthen the ability to establish a relationship between the open EC evaporation and pan evaporation at the daily scale, as discussed in Section 4.5."

p14 line 259;

"A sensitivity analysis of the daily pan evaporation and daily EC evaporation with air temperature, relative humidity, wind speed, and solar radiation, strengthen the ability to establish a relationship between the open EC evaporation and pan evaporation at the daily scale.

The $K_{pan}$ was parametrised as a function of the wind speed, air temperature, relative humidity, and solar radiation measured at Alquilha station. The $K_{pan}$ was 0.59, 0.57, 0.57, and 0.64 in June, July, August, and September, respectively..."

**- The factors governing evaporation – it needs more clarity on how the factors governing evaporation were determined. These factors are mentioned in the Methodology section, and form the base of the pan coefficient function that is developed. Are these governing factors identified based on literature or other results that are not shown here?**

Yes, the factors governing evaporation were identified mostly based on literature (see for instance Allen et al., 1998) but also, because they are the parameters measured in the Alquilha meteorological station.

We have provided more details for describing how the governing factors were determined in section 3:

p7 line 148;

"...First, relationships were determined between the EC measurements and meteorological parameters (air temperature, relative humidity, wind speed, and solar radiation) measured at Alqueva-Montante station. These four meteorological parameters were chosen mainly because, they are the factors governing evaporation usually describe in bibliography (see for instance Allen et al. 1998) and because they are the parameters measured in the Alquilha meteorological station. The daily cycle of evaporation and normalised meteorological parameters were analysed to assess their behaviour during the day. A sensitive analysis at the hourly scale was also performed for the factors governing evaporation from Alqueva Reservoir..."

**- The multivariable nonlinear pan coefficient function - could the authors explain how they came to the form of the multivariable nonlinear pan coefficient function, apart from the explanation that a linear function would not describe the correlation between EC evaporation and pan evaporation well.**

We add in section 1, some explanation and several bibliographic references which use the $K_{pan}$ as a function of meteorological parameters. In our case, we can say that, based on the four meteorological parameters measured at Alquilha station we try several functions and the best function (which leads to the minimum residual sum of squares and the better coefficient of determination) was the one that is presented in the paper. In this function, for instance, we take the logarithms of the radiation and the

relative humidity as the range of values of these two parameters is quite superior of the other two (temperature and wind speed), and when taking the logarithms, we can reduce the scale of the former parameters.

The text is now as follows:

p3 line 70;

"...The relationship between pan evaporation and lake evaporation must be a function of meteorological parameters. The pan evaporation method remains the cheapest and simplest method; hence, this evaporimeter remains the most commonly used instrument to quantify reservoir evaporation. The application of a pan coefficient to convert measured pan evaporation to reservoir evaporation is a method frequently applied in reservoir studies and this pan coefficient is often calculated as a function of meteorological parameters (Allen et al., 1998; Pereira et al., 1995; Pradhan et al., 2013)."

**- A clear description of the figures that are presented as results is sometimes lacking in my opinion. This is the case for figures 3, 7 and 9. What do we see in this figure, how do we read it, what is the main message that the reader can take from it? I think this will help your story to come across more direct and focussed, and will improve the guidance of the reader towards the conclusions that are well supported by the results.**

Regarding figure 9, we already add some text when responding to the second general comments, above. Regarding figure 3 and 7, we add some additional explanation to the manuscript in order to make those figures more clearly to the readers.

p5 line 140;

"...daily cycle with a $180°$ filter and without a filter ($R^2$ = 0.985) and between the daily cycle with a $100°$ filter and without a filter ($R^2$ = 0.993) are presented in Fig. 3b and

3c. Analysing these figures we can concluded that the platform does not affect the wind direction and based on this results, the wind direction filter applied in Potes et al. (2017) was not consider in this study."

p9 line 193;

The correlation between daily eddy covariance evaporation and daily pan evaporation were determined for the study period (June-September). In Fig. 7 these correlations are presented. Figure 7a shows a poor linear correlation between the EC evaporation and pan evaporation during the entire study period ($R^2$ = 0.37). This was also the case when observing the plots for each month: $R^2$ = 0.1882 in June, $R^2$ = 0.0458 in July, $R^2$ = 0.3345 in August, and $R^2$ = 0.4693 in September. These results show that the relation between the both evaporations could not be considered linear and reveal the importance of finding a nonlinear function to correlate EC evaporation and pan evaporation (Fig. 7b–e; $R^2$ = 0.05–0.47). These results reveal the importance of finding a multivariable nonlinear function to correlate EC evaporation and pan evaporation.

**- Another general comment that I would to make is to see if a better balance can be achieved between the size of the sections. Sections 3 and 4.4 are relatively short and misses information. Probably this can already be improved by applying the two comments mentioned above.**

Yes, when applying the comments mentioned above, we have re-written the section 3 and section 4.4, and consequently a better balance of the size of the sections were obtained.

**- The conclusions section somewhat misses a concluding statement and is now presented more as a summary. Furthermore, some new numbers are shown in**

**this section, which is not the appropriate place to present new results. Referring to p.13 l.253.**

Yes, we agree, and we have re-written the entire conclusions and the text is now as follows:

p12 line 242

The aim of this study was to develop a method to evaluate the evaporation from Alqueva Reservoir, south-east of Portugal, based on Class A pan measurements, thus providing an evaluation tool for water management within the Multipurpose Alqueva Project (MAP) and for other reservoirs with a Mediterranean climate.

Water fluxes were continuously measured from June to September 2014 by the EC method at Alqueva-Montante station to obtain accurate reservoir evaporation measurements. The total EC reservoir evaporation from June to September 2014 was 450.1 mm, and the mean daily evaporation in June, July, August, and September was 3.7 mm $d^{-1}$, 4.0 mm $d^{-1}$, 4.5 mm $d^{-1}$, and 2.5 mm $d^{-1}$, respectively. Considering the most important atmospheric factors controlling evaporation we observed, at the hourly scale, a positive correlation between the EC evaporation and wind speed and air temperature whereas a negative correlation was found for the relative humidity and no correlation between evaporation and solar radiation.

The Class A pan installed at Alquilha station provided hourly and daily pan evaporation values. The total pan evaporation from June to September 2014 was 797.9 mm, and the mean daily evaporation in June, July, August, and September was 6.9 mm $d^{-1}$, 7.7 mm $d^{-1}$, 7.3 mm $d^{-1}$, and 4.3 mm $d^{-1}$, respectively. Positive correlations were observed between the hourly pan evaporation and air temperature and solar radiation, whereas a negative correlation was found between the hourly pan evaporation and relative humidity. There was a weak correlation between the hourly pan evaporation and wind speed.

A sensitivity analysis of the daily pan evaporation and daily EC evaporation with air temperature, relative humidity, wind speed, and solar radiation, strengthen the ability to establish a relationship between the open EC evaporation and pan evaporation at the daily scale.

We found that the daily pan evaporation coefficient could be expressed by a multi-variable function of wind speed, air temperature, relative humidity, and solar radiation measured at Alquilha station, and the model validation were performed for the same four summer months in 2017. The modelled pan coefficients ($K_{pan}$) were 0.59, 0.57, 0.57, and 0.64 in June, July, August, and September, respectively. Consequently, the modelled daily mean $E_{Res}$ was 3.9 mm d$^{-1}$, 4.2 mm d$^{-1}$, 4.5 mm d$^{-1}$, and 2.7 mm d$^{-1}$ in June, July, August, and September, respectively. The total modelled evaporation was 455.8 mm, remarkably similar to total output from EC measurements, and corresponds to 101.3% of the measured EC evaporation from the reservoir.

The evaporation model proposed in this study can assist and improve water management in the MAP. Moreover, the methodology could also be applied to other reservoirs, and the equation developed for Alqueva Reservoir could act as a first evaluation for the management of other reservoirs in the region.

**Specific comments**

**- p.2 lines 46-48; how was the total reference evapotranspiration calculated? Using Penman-Monteith as mentioned at p.12 line 236?**

Yes. We add this information on the text:

"In the case of Alqueva Reservoir, with an average reference evapotranspiration of $\sim 1270$ mm per year (calculated by Penman-Montheith method), the evaporation can be..."

**- p.3 lines 77-82; I would like to suggest to describe at what timescales the study**

**focusses.**

The study is performed at daily scale. We add this information on the text:

"The study was undertaken using daily data for the period from June to September 2014, and was..."

**- p.5 lines 134/135; please check if the negative latent heat fluxes found are indeed erroneous, or is there condensation happening?**

Negative latent heat fluxes can be found in the Irgason system. As it is an open-path the water vapour concentration is obtained through infrared absorption in the optical path. Condensation in the optical windows can happen (that the system is able to reverse) and still the strength of the signal is within the acceptable range (0.7 - 1.0).

**- p.5 lines 137-141; it does not become clear how the authors have applied this filter. Does the wind direction filter have a range of 180o and 100o respectively, or is there a filter from 180o and 100o towards 360o? Please clarify from which to which wind direction the filter is applied.**

Yes, regarding the orientation of the anemometer, the wind direction filter has a range of $180°$ and $100°$, respectively. But, in this study no filter was applied as, now, explained, in the end of section 2.2.

**- p.7 lines 146/147; What conditions surrounding a site can influence the pan coefficient? Could the authors further explain if indeed those conditions can be ignored because the fetch in the wind direction was found not to be relevant.**

The condition that can influence the pan coefficient are, for instance: the ground cover in the station, its surroundings as well as the general wind and humidity conditions. (see for instance, Allen et al., 1998, pag.79) The Alquilha station is installed in an

island, located in the middle of the Alqueva reservoir. This island is small enough allowing to considerer that there is no influence of land in pan evaporation. In another words, we can considerer that the pan is surrounding by water.

**- p.7 line 161; How did the authors deal with the data that was filtered out in calculating the total evaporation amount?**

We have considered a period of 122 days (Jun-Sep) and to calculate the total evaporation amount we considered the average of the existing data multiplied by 122 days. In other words, we considered that value of the missing days was equal to the average value.

**- p.7 lines 173/174; The authors mention that the delay of evaporation is related to the variation in the energy storage in the water body, however this is not shown in figure 5. Do the authors have data on this that could be presented?**

No. In fact, the heat storage was not considered in this study, so we corrected the sentence to:

"Incoming solar radiation contributed to evaporation with a delay corresponding to that could be explained by the variation in the energy stored in the water column. "

**- p.7 lines 174/175; I think this argumentation could be written down more clearly. The increase in energy storage in the water body by solar radiation is not depending on the gradient of air-water temperature. The solar radiation will penetrate the water surface in any case.**

Yes, we re-written the sentence and add some references.

"Increased The increase solar radiation may lead to an increase in the stored energy in the water column (Potes et al, 2017, Nordbo et al, 2011) . when the air temperature was higher than the water temperature."

**- p.7 lines 176-178; at line 166 it is presented that there no correlation was found between open water evaporation and incoming solar radiation. However, in line 176-178 it is presented as if there is a direct correlation between the variables. Please elaborate.**

Yes, at hourly scale, there is no correlation between open water evaporation and incoming solar radiation (Fig. 4d) but when the mean daily cycle is analyzed it can be found a direct correlation. (Fig. 5).

**- p.10 lines 205-207; which results support this statement? As far as I can see there is no data presented on heat storage.**

As we mentioned above, the heat storage was not considered in this study, but we think that could be one of the explanations, so we corrected the sentence to:

"These results agree with a previous study by (Salgado and Le Moigne, 2010) for the same reservoir, whereby the authors evaporation measurement had similar times for their mean daily value (between 12:00 LT and 13:00 LT), the considerable dissimilarities over the day resulted from the large difference between the size of the pan and the size of the reservoir as these may lead to different heat storage capacities."

**- p. 12 lines 243/244; it would be interesting to know whether the method presented in this study can indeed be applied to other reservoirs with a Mediterranean climate. Could the authors discuss further on this?**

Yes, you are absolutely right. This study is focusses on only one reservoir, Alqueva reservoir, which is the largest reservoir in Portugal. We believe that Alqueva reservoir could represent quite well the conditions of most reservoir located in Mediterranean climate. We have conscience that furthers studies are needed but meanwhile the conclusion of this study could help water managers in reservoir evaporation calculation, as now they use a basic approximation of 1000 mm as the reservoir annual evaporation.

**Technical corrections**

**- p.2 line 28, 29; not sure if hm3 is a common unit to use. Consider changing.**

We change hm$^3$ to m$^3$.

**- p.2 line 31; (Kohli and Frenken, 2015) -> Kohli and Frenken (2015).**

Corrected as suggested.

**- p.2 lines 55/56; This sentence seems not in the right place in this location in the paragraph. Consider bringing it forward.**

Yes. The sentence was moved to the beginning of the paragraph:

"The turbulent fluxes over the water surface, which can be obtained with direct and continuous measurements, evaluate the exchange of water and energy between the surface and the atmosphere (Arya, 2001; Potes et al.,2017). The EC method is usually applied in research because it is a non-invasive technique and provides the most accurate and reliable method for estimating evaporation (Stull, 2001; Allen and Tasumi, 2005; Tanny et al., 2008; Rimmer et al., 2009). The method is theoretically based on the correlation between the vertical wind speed and air moisture content fluctuation, which is a reliable and accurate way to measure open-water evaporation in the location where it is installed (Blanken et al., 2000; Tanny et al., 2008; Nordbo et al., 2011; Richardson et al., 2012; Vesala et al., 2012; Liu et al., 2015; Ning et al., 2015; Ma et al., 2016). The turbulent fluxes over the water surface, which can be obtained with direct and continuous measurements, evaluate the exchange of water and energy between the surface and the atmosphere (Arya, 2001; Potes et al.,2017). However, it requires sophisticated instrumentation that is capable of accurately recording the minimum variations in wind speed, air temperature, and humidity with a high sampling frequency. Furthermore, the equipment is quite expensive and requires continuous maintenance, which means that is not possible to perform regular measurements..."

**- p.3 line 59; add 'it' to the sentence: 'which means that it is not possible. . .'**

Corrected as suggested.

**- p.3 line 64; waterbodies -> water bodies**

Corrected as suggested.

**- p.3 lines 83-85; This paragraph might be redundant. Especially mentioning about section 1, which the reader at that moment has just read.**

We eliminated the part referring to section 1 and he have re-written the paragraph:

Section 1 of this paper introduces the aims of the study, and The paper outline is as follows. Section 2 presents the measurement site, instrumentation, and data. The methodology used in this study is detailed in Section 3, and the results are presented and discussed in Section 4. Finally, Section 5 summarizes the major conclusions.

**- p.7 line 164 and other lines; trend should be correlation?**

Yes. We replaced trend for correlation whenever it applies.

**- p.7 line 166; open evaporation -> open water evaporation**

Corrected as suggested.

**- p.8 line 187; The most importance differences with what?**

Corrected in the next comment.

**- p.8 lines 187/188; The dominance of wind speed over solar radiation in relation to open water evaporation? Please clarify.**
What we want to say is that in the morning period the variable that most fit the evaporation curve is the wind speed. Nevertheless, for clarity sake we rewrite the sentence as follows:

"The daily cycle of evaporation and the four normalised meteorological parameters (wind speed, air temperature, relative humidity, and solar radiation) measured at Alquilha station are presented in Fig. 6. In the morning period, the solar radiation begins at 8:00 LT and with that an increase in air temperature and a decrease in relative humidity. At 11:00 LT wind speed starts to increase and around 12:00 LT occurs the trigger of the evaporation pan. The trend of the pan evaporation followed the trend of solar radiation but with a delay of about 3 hours, whereby the maximum value was at 16:00 LT when the relative humidity was at the minimum. Pan evaporation reduced as the air relative humidity increased. The most important differences that were observed are the dominance of the wind speed over solar radiation in the morning period (until 11:00 LT), even with the reduction of the relative humidity. When the wind speed increased, the trend of pan evaporation followed the trend of solar radiation but with a delay, whereby the maximum value was at 16:00 LT when the relative humidity was at the minimum. Pan evaporation reduced as the air relative humidity increased.

**- p.12 line 230; (Rodrigues, 2009) -> Rodrigues (2009).**

Corrected as suggested.

**- p.12 line 239; please clarify what is meant with 'high measured evaporation'? High evaporation rates? High measurement frequency?**

Yes, we mean evaporation rates, so we corrected the text to:

"The model's ability was tested for the period from June to September 2017 (Fig. 11; $R^2$ = 0.68); thus, the model could estimate the $E_{Res}$ despite the high measured evaporation rates and the reduced number of available daily pan evaporation measurements."

**- p.13 line 257; significative -> significant. Or should it be 'weak' instead of 'no significant' following from section 4.2.**

Yes, we mean weak correlation, so we corrected the text to:

"...There was no significative trend a weak correlation between the hourly pan evaporation and wind speed."

**References**

Pereira, R., Nova, N., Pereira, A. and Barbieri, V.: A model for the class A pan coefficient, Agric. For. Meteorol., 76(2): 75-82, doi: doi.org/10.1016/0168-1923(94)02224-8, 1995.

Pradhan, S., Sehgal, B., Das, D., Bandyopadhyay and Singh, R.: Evaluation of pan coefficient methods for estimating FAO-56 reference crop evapotranspiration in a semi-arid environment, Journal of Agrometeorology, 15(1), 90-93, https://www.researchgate.net/publication/291765647, 2013.

World Meteorological Organization: Guide to Instruments and Methods of Observation, WMO N° 8, Volume I – Measurement of Meteorological Variables. Geneva, 2018.

---

## Referee Comment (RC2) · Anonymous Referee #2 · 26 Sep 2020

The authors analyze the relationship between Pan evaporation and EC evaporation at Alqueva Reservoir. The modelled pan coefficient was estimated to be 0.59, 0.57, 0.57, and 0.64 in June, July, August, and September, respectively. The developed pan coefficient function was further validated for the same period in 2017. This study is useful to estimate evaporation at Alqueva Reservoir based on pan measurement. As far as my knowledge, I think the submission is worthy of being published in HESS after a minor revision. Some major comments are listed as the following: Line 10-12, What is the difference of EC evaporation and modeled evaporation? Same to Line 15. Line 28 and line 90, hm and ha are not common units. Line 70, Why the relationship between pan evaporation and lake evaporation must be a function of meteorological

parameters? In fact, lake heat storage is also a main factor of the difference between pan evaporation and lake evaporation Line 81, Can the pan coefficient function in June to September is be used to other months? Line 144-145, What is the theoretical basis? Line 150-156, The expression is not clear enough, please address it in more detail. Fig.8, it is difficult to differentiate the two curves. Please change the color. Section 4.4 is two simple and should be addressed in more detailed.

---

## Author Comment (AC2) · 3 Oct 2020

[hess,manuscript]copernicus

Manuscript reference No. HESS-2020-283

We would like to thank you, for your insightful comments, which unquestionably contributed to improve our manuscript. We believe that we were able to fully and adequately respond and address all your questions and recommendations. In the following pages are our point-by-point responses to each of your comments as well as your own

comments.

Revisions in the text are shown using green colour font for [example] additions , and strike through red font [example] for deletions.

**Line 10-12 - What is the difference of EC evaporation and modeled evaporation? Same to Line 15.**

The daily mean reservoir evaporation (EC) was measured in the lake, by the IRGASON, and the modelled evaporation ($E_{Res}$) was obtained by the pan evaporation method, where the $K_{pan}$ was modelled as a function of the four meteorological parameters.

**Line 28 and line 90, hm and ha are not common units.**

We have changed hm$^3$ to m$^3$, and ha to km$^2$.

**Line 70, Why the relationship between pan evaporation and lake evaporation must be a function of meteorological parameters? In fact, lake heat storage is also a main factor of the difference between pan evaporation and lake evaporation.**

Yes, we agree that the sentence is not clear, thus we re-written as:

"It is expected that the relationship between pan evaporation and lake evaporation should must be a function of meteorological parameters, through the modelled Kpan."

**Line 81, Can the pan coefficient function in June to September is be used to other months?**

No, this study was developed for the summer months and cannot be used to other months. These months was chosen because they represent about 60% of the total
reference evapotranspiration in a Mediterranean climate. This was already referred in line 46-49.

**Line 144-145, What is the theoretical basis?**

The theoretical basis is described by several authors. We added a reference in the end of the sentence:

It is proposed that the actual evaporation from the reservoir could be estimated using the relationship between the Class A pan evaporation measurements (at Alquilha station) and a pan coefficient multivariable function, as determined by Allen et al., (1998) but for reference evapotranspiration.

Also, the following sentence was added in the Section 1, Line 72:

"...the most commonly used instrument to quantify reservoir evaporation. The application of a pan coefficient to convert measured pan evaporation to reservoir evaporation is a method frequently applied in reservoir studies and this pan coefficient is often calculated as a function of meteorological parameters (Allen et al., 1998; Pereira et al., 1995; Pradhan et al., 2013)."

**Line 150-156, The expression is not clear enough, please address it in more detail.**

We agree with the reviewer. We have re-written a major part of Section 3:

"...First, relationships were determined between the EC measurements and meteorological parameters (air temperature, relative humidity, wind speed, and solar radiation) measured at Alqueva-Montante station. These four meteorological parameters were chosen mainly because, they are the factors governing evaporation usually describe in bibliography (see for instance Allen et al., 1998) and because they are the parameters measured in the Alquilha meteorological station. The daily cycle of evaporation

and normalised meteorological parameters were analysed to assess their behaviour during the day. A sensitive analysis at the hourly scale was 160 also performed for the factors governing evaporation from Alqueva Reservoir. Second, the relationships were determined between pan evaporation measurements and the same meteorological parameters, but as measured at Alquilha station (at hourly and daily scales).

Third the correlation between EC evaporation and pan evaporation where determined and the daily cycles of the normalized pan evaporation and normalised EC evaporation are compared.

Forth a sensitivity analysis of pan evaporation and EC evaporation versus meteorological variables was performed.

Fifth, the daily multivariable pan coefficient series was calculated, by dividing the daily values of EC evaporation by the daily values of pan evaporation.

Sixth, a function was fitted to this series based on the physical relationship between the meteorological parameters measured at Alquilha station (at the daily scale). Several functions were attempted, and the one leading to a better determination coefficient ($R^2$) was chosen. In order to find the optimal parameter estimates, the Generalized Reduced Gradient (GRG) method (Lasdon et al., 1974) was used with the aid of the Excel solver tool. The best parameter estimates were those that minimized the residual sum of squares."

**Fig.8, it is difficult to differentiate the two curves.**

Yes, we changed the colors to make it clearer.

**Section 4.4 is two simple and should be addressed in more detailed.**

Yes, we agree. We have re-written the entire section:

"**4.4 Sensitivity analysis of pan evaporation and EC evaporation versus meteoro-**

**logical variables**

A sensitivity analysis of the daily pan evaporation and daily EC evaporation with air temperature, relative humidity, wind speed, and solar radiation, was carried out and . T the results are presented in Fig.9 , . Fig. 9a show a non-linear correlation between evaporation (EC and pan evaporation) with wind speed. It can be seen that both evaporations have a positive linear correlation with air temperature, Fig. 9b, and radiation, Fig. 9d. In Fig 9c it can be seen a negative correlation between evaporation and air relative humidity. The value of R2 of pan evaporation with air temperature, air relative humidity and radiation is greater than the R2 of the EC evaporation with the same parameters. On the contrary the R2 of EC evaporation with wind speed is greater than the pan evaporation with the wind speed parameter. Based on this sensitivity analysis, the four parameters appear to cause influence in both EC evaporation and pan evaporation, and strengthen the ability to establish a relationship between the open EC evaporation and pan evaporation at the daily scale as discussed in Section 4.5."

---

## Author Response (AR1)

Ryan Teuling
Editor
Hydrology and Earth System Sciences
October 25th 2020

Manuscript reference No. HESS-2020-283

Dear Ryan Teuling,

10    Please find attached a revised version of our manuscript "Reservoir evaporation in a Mediterranean climate: Comparing direct methods in Alqueva Reservoir, Portugal".

In the revised manuscript we included the researcher Miguel Potes, as co-author. The reason for the inclusion of the Researcher Miguel Potes to the author's list is justified by the work he performed in installation, maintenance and data treatment of the Eddy-Covariance system. As well as in the contribution to analysis and interpretation of the results.

15    We would like to thank the reviewers for their insightful comments, which unquestionably contributed to improve our manuscript. We believe that we were able to fully and adequately respond and address all their questions and recommendations by re-writing important sections of the manuscript.

We hope that these improvements will now render our manuscript acceptable for publication as a Research Paper in Hydrology and Earth System Sciences.

Revisions in the text are shown using green colour font for [example] additions , and strike through red font [example] for deletions.

**List of all relevant changes made in the manuscript**
    - We have re-written the Abstract;
25    - We have rearranged Figures 3 and 7;
    - We have edited the text for language, grammar, and improved clarity.

    - In accordance with reviewer 1 suggestion:
    (1) We have provided more details for description the quality control process;
    (2) We add some additional text to explain more clearly the sensitivity analysis in section 4.4, in Methodology and in
30  Conclusions;
    (3) We have provided more details for describing how the governing factors were determined, in section 3;
    (4) We add in section 1, some explanation and several bibliographic references which use the $K_{pan}$ as a function of meteorological parameters;
    (5) We have re-written the entire conclusions.

35    - In accordance with reviewer 2 suggestion:
    (1) We have changed hm$^3$ to m$^3$ and ha to km$^2$;
    (2) We have re-written a major part of Section 3;
    (2) We have re-written Section 4.4.

**Responses to the comments of Reviewer 1**
40

**General comments**
**- The quality control process of the Class A pan in section 2.2. Please elaborate on what method this quality control process is based on**.

45    The quality control process is based on the analysis of the existing data in order to discard the values that, for any reasons, could not be considered adequate. Following, we have discarded:

- The values obtained 3 hours after each refill of the pan;
- The values obtained when the water level in the pan is below a threshold value (10 cm), according to Allen et al., 1998 and WMO, 2018;
50 - The anomalous values.

We have provided more details for description the quality control process in section 2.2.

**- The sensitivity analysis - I would like to read more on how the authors have performed the sensitivity analysis. This does not become clear from the Methodology section, nor from the results in section 4.4.**

55 The sensitivity analysis was done by determining the correlation between evaporation (daily pan evaporation and daily EC evaporation) and the four meteorological parameters measured at Alquilha station (because this station will be used to obtain data in the future).

We add some additional text to the manuscript in order to explain more clearly the sensitivity analysis in section 4.4. We add also some text in Methodology and in Conclusions.

60 **- The factors governing evaporation – it needs more clarity on how the factors governing evaporation were determined. These factors are mentioned in the Methodology section, and form the base of the pan coefficient function that is developed. Are these governing factors identified based on literature or other results that are not shown here?**

Yes, the factors governing evaporation were identified mostly based on literature (see for instance Allen et al., 1998) but also, because they are the parameters measured in the Alquilha meteorological station.
65 We have provided more details for describing how the governing factors were determined, in section 3.

**- The multivariable nonlinear pan coefficient function - could the authors explain how they came to the form of the multivariable nonlinear pan coefficient function, apart from the explanation that a linear function would not describe the correlation between EC evaporation and pan evaporation well.**

We add in section 1, some explanation and several bibliographic references which use the $K_{pan}$ as a function of meteoro-
70 logical parameters. In our case, we can say that, based on the four meteorological parameters measured at Alquilha station we try several functions and the best function (which leads to the minimum residual sum of squares and the better coefficient of determination) was the one that is presented in the paper. In this function, for instance, we take the logarithms of the radiation and the relative humidity as the range of values of these two parameters is quite superior of the other two (temperature and wind speed), and when taking the logarithms, we can reduce the scale of the former parameters.

75 **- A clear description of the figures that are presented as results is sometimes lacking in my opinion. This is the case for figures 3, 7 and 9. What do we see in this figure, how do we read it, what is the main message that the reader can take from it? I think this will help your story to come across more direct and focussed, and will improve the guidance of the reader towards the conclusions that are well supported by the results.**

Regarding figure 9, we already add some text when responding to the second general comments, above. Regarding figure 3
80 and 7, we add some additional explanation to the manuscript in order to make those figures more clearly to the readers.

**- Another general comment that I would to make is to see if a better balance can be achieved between the size of the sections. Sections 3 and 4.4 are relatively short and misses information. Probably this can already be improved by applying the two comments mentioned above.**

Yes, when applying the comments mentioned above, we have re-written the section 3 and section 4.4, and consequently a
85 better balance of the size of the sections were obtained.

**- The conclusions section somewhat misses a concluding statement and is now presented more as a summary. Furthermore, some new numbers are shown in this section, which is not the appropriate place to present new results. Referring to p.13 l.253.**

Yes, we agree, and we have re-written the entire conclusions.

**Specific comments**

**- p.2 lines 46-48; how was the total reference evapotranspiration calculated? Using Penman-Monteith as mentioned at p.12 line 236?**

Yes. We add this information on the text:

"In the case of Alqueva Reservoir, with an average reference evapotranspiration of $\sim$ 1270 mm per year (calculated by the Penman-Montheith method), the evaporation can be..."

**- p.3 lines 77-82; I would like to suggest to describe at what timescales the study focusses.**

The study is performed at daily scale. We add this information on the text:

"The study use daily data for the period from June to September 2014, and was..."

**- p.5 lines 134/135; please check if the negative latent heat fluxes found are indeed erroneous, or is there condensation happening?**

Negative latent heat fluxes can be found in the Irgason system. As it is an open-path the water vapour concentration is obtained through infrared absorption in the optical path. Condensation in the optical windows can happen (that the system is able to reverse) and still the strength of the signal is within the acceptable range (0.7 - 1.0).

**- p.5 lines 137-141; it does not become clear how the authors have applied this filter. Does the wind direction filter have a range of 180º and 100º respectively, or is there a filter from 180º and 100º towards 360º? Please clarify from which to which wind direction the filter is applied.**

Yes, regarding the orientation of the anemometer, the wind direction filter has a range of 180º and 100º, respectively. But, in this study no filter was applied as, now, explained, in the end of section 2.2.

**- p.7 lines 146/147; What conditions surrounding a site can influence the pan coefficient? Could the authors further explain if indeed those conditions can be ignored because the fetch in the wind direction was found not to be relevant.**

The condition that can influence the pan coefficient are, for instance: the ground cover in the station, its surroundings as well as the general wind and humidity conditions. (see for instance, Allen et al., 1998, p.79).

The Alquilha station is installed in an island, located in the middle of the Alqueva reservoir. This island is small enough allowing to considerer that there is no influence of land in pan evaporation. In another words, we can considerer that the pan is surrounding by water.

**- p.7 line 161; How did the authors deal with the data that was filtered out in calculating the total evaporation amount?**

We have considered a period of 122 days (Jun-Sep) and to calculate the total evaporation amount we considered the average of the existing data multiplied by 122 days. In other words, we considered that value of the missing days was equal to the average value.

**- p.7 lines 173/174; The authors mention that the delay of evaporation is related to the variation in the energy storage in the water body, however this is not shown in figure 5. Do the authors have data on this that could be presented?**

No. In fact, the heat storage was not considered in this study, so we corrected the sentence to:

"Incoming solar radiation contributed to evaporation with a delay  that could be explained by the variation in the energy stored in the water column. "

**- p.7 lines 174/175; I think this argumentation could be written down more clearly. The increase in energy storage in the water body by solar radiation is not depending on the gradient of air-water temperature. The solar radiation will penetrate the water surface in any case.**

Yes, we re-written the sentence and add some references.

130 " The increase in solar radiation may lead to an increase in the stored energy in the water column (Potes et al, 2017, Nordbo et al, 2011) . "

**- p.7 lines 176-178; at line 166 it is presented that there no correlation was found between open water evaporation and incoming solar radiation. However, in line 176-178 it is presented as if there is a direct correlation between the variables. Please elaborate.**

135 Yes, at hourly scale, there is no correlation between open water evaporation and incoming solar radiation (Fig. 4d) but when the mean daily cycle is analyzed it can be found a direct correlation. (Fig. 5).

**- p.10 lines 205-207; which results support this statement? As far as I can see there is no data presented on heat storage.**

As we mentioned above, the heat storage was not considered in this study, but we think that could be one of the explanations,
140 so we corrected the sentence to:

"These results agree with a previous study by (Salgado and Le Moigne, 2010) for the same reservoir, wherein the authors observed an absolute minimum and maximum at 6:00 LT and 21:00 LT, respectively. Although both types of evaporation measurement had similar times for their mean daily value (between 12:00 LT and 13:00 LT), the considerable dissimilarities over the day resulted from the large difference between the size of the pan and the size of the reservoir as these may lead to
145 different heat storage capacities."

**- p. 12 lines 243/244; it would be interesting to know whether the method presented in this study can indeed be applied to other reservoirs with a Mediterranean climate. Could the authors discuss further on this?**

Yes, you are absolutely right. This study is focusses on only one reservoir, Alqueva Reservoir, which is the largest reservoir in Portugal. We believe that Alqueva Reservoir could represent quite well the conditions of most reservoir located in
150 Mediterranean climate. We have conscience that furthers studies are needed but meanwhile the conclusion of this study could help water managers in reservoir evaporation calculation, as now they use a basic approximation of 1000 mm as the reservoir annual evaporation.

**Technical corrections**

**- p.2 line 28, 29; not sure if hm3 is a common unit to use. Consider changing.**
155 We change hm$^3$ to m$^3$.

**- p.2 line 31; (Kohli and Frenken, 2015) -> Kohli and Frenken (2015).**
Corrected as suggested.

**- p.2 lines 55/56; This sentence seems not in the right place in this location in the paragraph. Consider bringing it forward.**
160 Yes. The sentence was moved to the beginning of the paragraph:

"The turbulent fluxes over the water surface, which can be obtained with direct and continuous measurements, evaluate the exchange of water and energy between the surface and the atmosphere (Arya, 2001; Potes et al., 2017). The EC method is usually applied in research because it is a non-invasive technique and provides the most accurate and reliable method for estimating evaporation (Stull, 2001; Allen and Tasumi, 2005; Tanny et al., 2008; Rimmer et al., 2009). The method is
165 theoretically based on the correlation between the vertical wind speed and air moisture content fluctuation and is a reliable and accurate method to measure open-water evaporation in a location where it is installed (Blanken et al., 2000; Tanny et al., 2008; Nordbo et al., 2011; Richardson et al., 2012; Vesala et al., 2012; Liu et al., 2015; Ning et al., 2015; Ma et al., 2016).

of water and energy between the surface and the atmosphere (Arya, 2001; Potes et al., 2017). However, it requires sophisticated
170 instrumentation that is capable of accurately recording the minimum variations in wind speed, air temperature, and humidity
with a high sampling frequency. Furthermore, the equipment is quite expensive and requires continuous maintenance, which
means that it is not possible to perform regular measurements..."

**- p.3 line 59; add 'it' to the sentence: 'which means that it is not possible...'**
Corrected as suggested.

175 **- p.3 line 64; waterbodies -> water bodies**
Corrected as suggested.

**- p.3 lines 83-85; This paragraph might be redundant. Especially mentioning about section 1, which the reader at that moment has just read.**
We eliminated the part referring to section 1 and he have re-written the paragraph:
180 Section 1 of this paper introduces the aims of the study, and The paper is organised as follows. Section 2 presents the
measurement site, instrumentation, and data. The methodology used in this study is detailed in Section 3, and the results are
presented and discussed in Section 4. Finally, Section 5 summarizes the major conclusions.

**- p.7 line 164 and other lines; trend should be correlation?**
Yes. We replaced trend for correlation whenever it applies.

185 **- p.7 line 166; open evaporation -> open water evaporation**
Corrected as suggested.

**- p.8 line 187; The most importance differences with what?**
Corrected in the next comment.

**- p.8 lines 187/188; The dominance of wind speed over solar radiation in relation to open water evaporation? Please**
190 **clarify.**

What we want to say is that in the morning period the variable that most fit the evaporation curve is the wind speed.
Nevertheless, for clarity sake we rewrite the sentence as follows:
"The daily cycle of evaporation and the four normalised meteorological parameters (wind speed, air temperature, relative
humidity, and solar radiation) measured at Alquilha station are presented in Fig. 6. In the morning period, the solar radiation
195 begins at 8:00 LT and with that an increase in air temperature and a decrease in relative humidity. At 11:00 LT wind speed
starts to increase and around 12:00 LT occurs the trigger of the evaporation pan. The trend of the pan evaporation followed
the trend of solar radiation but with a delay of about 3 hours, whereby the maximum value was at 16:00 LT when the relative
humidity was at the minimum. Pan evaporation reduced as the air relative humidity increased. The most important differences
that were observed are the dominance of the wind speed over solar radiation in the morning period (until 11:00 LT), even with
200 the reduction of the relative humidity. When the wind speed increased, the trend of pan evaporation followed the trend of solar
radiation but with a delay, whereby the maximum value was at 16:00 LT when the relative humidity was at the minimum. Pan
evaporation reduced as the air relative humidity increased."

**- p.12 line 230; (Rodrigues, 2009) -> Rodrigues (2009).**
Corrected as suggested.

205 **- p.12 line 239; please clarify what is meant with 'high measured evaporation'? High evaporation rates? High mea-**
**surement frequency?**
Yes, we mean evaporation rates, so we corrected the text to:

"...the model could estimate the $E_{Res}$ despite the high measured evaporation rates and the reduced number of available daily pan evaporation measurements."

210  **- p.13 line 257; significative -> significant. Or should it be 'weak' instead of 'no significant' following from section 4.2.**
Yes, we mean weak correlation, so we corrected that in the text.

**Responses to the comments of Reviewer 2**

**- Line 10-12 - What is the difference of EC evaporation and modeled evaporation? Same to Line 15.**
215  The daily mean reservoir evaporation (EC) was measured in the lake, by the IRGASON, and the modelled evaporation ($E_{Res}$) was obtained by the pan evaporation method, where the $K_{pan}$ was modelled as a function of the four meteorological parameters.

**- Line 28 and line 90, hm and ha are not common units.**
We have changed $hm^3$ to $m^3$, and ha to $km^2$.

220  **- Line 70, Why the relationship between pan evaporation and lake evaporation must be a function of meteorological parameters? In fact, lake heat storage is also a main factor of the difference between pan evaporation and lake evaporation.**
Yes, we agree that the sentence is not clear, thus we re-written as:
"It is expected that the relationship between pan evaporation and lake evaporation should  be a function of meteorolog-
225  ical parameters, through the modelled $K_{pan}$."

**- Line 81, Can the pan coefficient function in June to September is be used to other months?**
No, this study was developed for the summer months and cannot be used to other months. These months was chosen because they represent about 60% of the total reference evapotranspiration in a Mediterranean climate. This was already referred in line 46-49.

230  **- Line 144-145, What is the theoretical basis?**
The theoretical basis is described by several authors. We added a reference in the end of the sentence:
It is proposed that the actual evaporation from the reservoir could be estimated using the relationship between the Class A pan evaporation measurements (at Alquilha station) and a pan coefficient multivariable function, as determined by Allen et al., (1998) but for reference evapotranspiration.
235  Also, the following sentence was added in the Section 1, Line 72:
"...the most commonly used instrument to quantify reservoir evaporation. The application of a pan coefficient to convert measured pan evaporation to reservoir evaporation is a method frequently applied in reservoir studies and this pan coefficient could be calculated as a function of meteorological parameters (Allen et al., 1998; Pereira et al., 1995; Pradhan et al., 2013)."

**- Line 150-156, The expression is not clear enough, please address it in more detail.**
240  We agree with the reviewer. We have re-written a major part of Section 3

**- Fig.8, it is difficult to differentiate the two curves.**
Yes, we changed the colors to make it clearer.

**- Section 4.4 is two simple and should be addressed in more detailed.**
Yes, we agree. We have re-written the entire section.

[revised manuscript text omitted]
 was 3.7 mm d$^{-1}$, 4.0 mm d$^{-1}$, 4.5 mm d$^{-1}$, and 2.5 mm d$^{-1}$, respectively. At the hourly scale, positive trend were observed between the EC evaporation and i) wind speed (R$^2$ = 0.50) and ii) air temperature (R$^2$ = 0.20), whereas a negative trend was found between open evaporation and relative humidity (R$^2$ = 0.30). There was no trend between open evaporation and incoming solar radiation.~~

~~The Class A pan installed at Alquilha station provided hourly and daily pan evaporation values. As result of the quality control process, 18% and 15% of the data were omitted at hourly and daily scale, respectively. 
[revised manuscript text omitted]

---

## Author Response (AR2)

Ryan Teuling Editor Hydrology and Earth System Sciences November 5th 2020

5

Manuscript reference No. HESS-2020-283

Dear Ryan Teuling,

10 Please find attached a revised version of our manuscript "Reservoir evaporation in a Mediterranean climate: Comparing direct methods in Alqueva Reservoir, Portugal".

We would like to thank the reviewer, Femke Jense, for her comments which, again, contributed to improve our manuscript. We hope that these improvements will now render our manuscript acceptable for publication as a Research Paper in Hydrology and Earth System Sciences.

15

30

Revisions in the text are shown using green colour font for [example] additions, and strike through red font [example] for deletions.

**Responses to Report #1 by Femke Jense:**

- please improve the description of what method you used to perfrom a sensitivity analysis in the Methods section (p.7,l.167).

We have improved the description of the sensitivity analysis in the Methodology section.

**I still think that explaining the wind direction filter can use some improvement (p.6, l.146). In the response of the authors to my feedback they mention that it is a range of 180 degrees and a range of 100 degrees. I understand that. But starting at which wind direction? Is it a filter from 0 to 180 degrees for instance?**

In the filter of 180 degrees it is a filter of wind directions between 90 and 270 degrees, these wind directions represent winds that pass through the platform before reaching the EC instrument. In the filter of 100 degrees it is a filter between 130 and 230 degrees which represents a smaller area but in these cases the possible contribution of the platform is higher.

We have provided more details for description the the wind direction filter.

- After my question about how the authors dealt with gap-filling, they explained it well. I would like suggest to the authors to consider to implement this into the Methods section.

The explanation of gap-filling was introduced in Section 2.2 (Instrumentation, data sources, and quality) when we explained the data processing.

[revised manuscript text omitted]